# A 3D Composited Flexible Sensor Based on Percolative Nanoparticle Arrays to Discriminate Coupled Pressure and Strain

**DOI:** 10.3390/s23135956

**Published:** 2023-06-27

**Authors:** Linqi Ye, Xinlei Li, Xinle Yi, Pan Tang, Minrui Chen

**Affiliations:** College of Chemical Engineering, Zhejiang University of Technology, Hangzhou 310014, China; 2112001036@zjut.edu.cn (L.Y.); 2112101371@zjut.edu.cn (X.L.); 2112101027@zjut.edu.cn (X.Y.); 2112101037@zjut.edu.cn (P.T.)

**Keywords:** percolative nanoparticle arrays, 3D composited flexible substrate, multifunctional sensor, mechanical interference, quantum transport

## Abstract

Flexible mechanical sensors based on nanomaterials operate on a deformation-response mechanism, making it challenging to discern different types of mechanical stimuli such as pressure and strain. Therefore, these sensors are susceptible to significant mechanical interference. Here, we introduce a multifunctional flexible sensor capable of discriminating coupled pressure and strain without cross-interference. Our design involves an elastic cantilever fixed on the pillar of the flexible main substrate, creating a three-dimensional (3D) substrate, and two percolative nanoparticle (NP) arrays are deposited on the cantilever and main substrate, respectively, as the sensing materials. The 3D flexible substrate could confine pressure/strain loading exclusively on the cantilever or main substrate, resulting in independent responses of the two nanoparticle arrays with no cross-interference. Benefitting from the quantum transport in nanoparticle arrays, our sensors demonstrate an exceptional sensitivity, enabling discrimination of subtle strains down to 1.34 × 10^−4^. Furthermore, the suspended cantilever with one movable end can enhance the pressure perception of the NP array, exhibiting a high sensitivity of −0.223 kPa^−1^ and an ultrahigh resolution of 4.24 Pa. This flexible sensor with multifunctional design will provide inspiration for the development of flexible mechanical sensors and the advancement of decoupling strategies.

## 1. Introduction

Mechanical flexible sensors can be integrated into micromechanical electronic systems (MEMS) [1,2,3], enabling them to detect mechanical stimuli including strain [4], pressure [5], force [6], and vibration [7]. These sensors, which are considered as the essential building blocks for the future generation of intelligent systems, are currently widely applied in the automotive industry, health care technology, and clinical medicine, due to the advantages of flexibility, portability, and lightweight [6,8]. Consequently, there has been a significant surge of interest among researchers toward the development of flexible mechanical sensors with high performance in recent years.

Most flexible mechanical sensors are composed of elastic substrates and conducting sensing nanomaterials [9,10,11,12,13]. For example, Lee et al. prepared porous nanofibrous thin layers of carbon nanotubes, graphene and fluorinated copolymer by electrospinning [14]. The numerous internal pores inside the materials can be compressed by normal pressure to enhance the intercontact of the conducting nanomaterials, thus enabling the sensing materials to detect tiny pressure loads with an ultrahigh change factor over 1 × 10^6^. In another study, Liu et al. fabricated piezoresistive composite films by incorporating silver nanowires into an ethylene-co-vinyl acetate matrix through solvodynamic printing [15]. These sensing materials demonstrated the ability to detect pressures as low as 12.9 Pa, with the highest sensitivity recorded at 41.8 kPa^−1^. Boland et al. added graphene nanosheets to a cross-linked polysilicone, and found these conductive nanocomposites are sensitive electromechanical sensors with gauge factors >500 that can measure even the impact associated with the footsteps of a small spider [13]. It is easy to find that these mechanical sensing materials all seem to follow a common response mechanism. That is, when mechanical stimuli are applied to these sensors, the sensing materials undergo deformation, causing a change in the conductivity of the internal conducting networks. As a result, the internal resistance or capacitance of the sensing materials change with mechanical stimuli loading [9,10,12,13]. However, according to this operating mechanism, these mechanical sensors may exhibit a similar response to various kinds of mechanical stimuli, making it difficult to discern the type of mechanical stimuli accurately [14,16]. For example, the porous nanofibrous pressure sensors introduced above are still difficult to be completely insensitive to bending [14], and the piezoresistive composite films are also sensitive to strain [15]. Therefore, unless applied in situations only involving one specific type of mechanical stimuli, current mechanical flexible sensors suffer from cross-interference from other mechanical stimuli when applied in complex scenarios. It is still challenging to design flexible sensors based on conductive nanomaterials with the capability of distinguishing complex mechanical stimuli with high performance [17].

Recently, some conducting nanoparticles (NPs) are assembled as percolative dense arrays to construct flexible sensing materials [18,19]. In these NP arrays, the electronic transport involving tunneling and hopping, is sensitive to the deformation of the sensing array, resulting in NP array-based sensors with the capability of detecting mechanical stimuli with ultrahigh sensitivities and resolutions as well as ultralow power consumption [20,21]. It is worth noting that the properties of sensors could be tailored by adjusting the size of particles, the array coverage, the deposition position, and even the geometric dimension of substrates. The high-level designability of NP array-based sensing materials opens up great possibilities for fabricating multifunctional flexible sensors capable of discriminating the coupled mechanical stimuli [22]. However, the question of how to transform the high-level designability of NP arrays into flexible sensors that can discern pressure and strain simultaneously without interference still has to be investigated.

In fact, some three-dimensional (3D) elastic cantilever structures have been applied to develop many kinds of sensors with excellent sensing performance [23,24,25]. In addition, the 3D sensing structures can break through the dimensional limitations of traditional planar designs by confining the effects of different stimuli to different planes. Hence, we present a multifunctional flexible sensor to differentiate the in situ coupled pressure and strain. This 3D flexible sensors involves an elastic cantilever fixed on the pillar of the flexible main substrate, and two percolative nanoparticle (NP) arrays deposited on the cantilever and main substrate, respectively. Based on this specific sensing structure, the external pressure is loaded on the cantilever and the internal strain occurs on the main substrate, so that the cross-interference that the current mechanical sensors are susceptible to could be eliminated sufficiently. The sensor demonstrates excellent sensing characteristics for both pressure and strain. In particular, when responding to pressure, the sensor presented an impressive performance with ultrahigh sensitivity and minimal hysteresis, because of the mechanical sensing cantilever with only one end fixed and the other end free to move. This advantageous design allows the movable end to easily undergo buckling [26], as it primarily overcomes a bending resistance rather than an elastic resistance [27], to minimize the occurrence of significant plastic deformation on the cantilever [28]. This multifunctional mechanical sensor, which can sense pressure and strain simultaneously, is expected to find wide applications in accurately detecting complex stimuli.

## 2. Materials and Methods

### 2.1. Fabrication of 3D NP Array-Based Sensors

In our NP array-based multifunctional design, we chosen two flexible elastomers, polyethylene terephthalate (PET) and polydimethylsiloxane (PDMS), as the substrates. PET cantilevers of varied thicknesses and excellent quality were procured straight from the market, while the PDMS main substrates were produced by pouring a 10:1 mixture of elastomer of Sylgard 184, Dow Corning into a polytetrafluoroethylene (PTFE) mold with a square hole and curing. Thus, we could obtain a PDMS substrate with a 6 mm high pillar after peering off. We used silver (Ag) fuel with high conductivity to print electrodes. The shadow-mask evaporation was employed to print two silver interdigital electrodes (IDEs) with separations of 100 μm on the upper surface of PET and the lower surface of PDMS [29]. Figure 1a shows the structure of our NP array-based multifunctional sensor. Numerous metals, including gold (Au), tin (Sn), and zinc (Zn), could be used to create NP arrays. In this paper, palladium (Pd) was used to create NP arrays because of its exceptional chemical stability [30]. The Pd NP arrays were generated from the gas aggregation source and deposited into the IDEs on both substrates [20,31], to fabricate the sensing elements (SEs) on the PET and PDMS. The schematic of the NPs deposition system is shown in Figure 1b. The conductance evolution of assembled NP arrays could be monitored during deposition to control the percolation of NP arrays. Inset of Figure 1b depicts a typical nanoparticle deposition percolation curve. It is unable to detect any conductance of NP arrays at this early stage of the deposition, because the particles in IDEs are still too less to establish a conducting pathway. With continued deposition, the nanoparticles are enough to form the least conducting pathways which could be discerned by our instrument, thus forming a percolative threshold in the evolution curve. As more nanoparticles reach the surface, the complex pathways are interconnected into a network structure, and the conductance rises rapidly until we stop depositing when the conductance of the NP array exceeds 0.5 μS. In fact, the continuous deposition leads subsequent NPs to fill the insulating separation in the NP arrays, resulting in a gradual transformation of electronic transport from quantized tunneling/hopping to the classical flowing, which may degrade the sensitivities of the sensing elements to mechanical stimuli. Controlling the amount of NP deposition to ensure the quantized transport domaining the transport behavior in NP arrays, is much significant for the performance guarantee of the sensors [20]. We purchased silver particles and palladium target materials with a purity of 99.99% to vapor-deposit IDEs and deposit NP arrays, respectively.

### 2.2. Finite Element Analysis (FEA) of NP Array-Based Flexible Multifunctional Sensor

FEA is performed with the commercial software Comsol Multiphysics 5.6 [32,33,34], to simulate the deformations on the substrates under the action of different stimuli. In simulations, the NP array-based flexible multifunctional sensor consists of a PDMS film measuring 20 × 20 × 0.2 mm^3^ with a 6 mm high pillar at its center. Attached to top of the pillar is a PET cantilever measuring 15 × 5 × 0.1 mm^3^. They are set with the below properties: the density *ρ* = 1380 kg/m^3^ for PET and 970 kg/m^3^ for PDMS, the Young Modulus *E* = 2.8 GPa of PET and 750 kPa of PDMS, and the Poisson’s ratio *ν* = 0.3 of PET and 0.49 of PDMS. For the simulation of pressure loading, one end of the PET cantilever is fixed, and a force of 0.01 N is loaded on the edge at the movable end with a direction at an angle of 90° to the PET plane. The simulation of the surface deformation of the PET under force/pressure is presented in Figure 2(ai). For the simulation of strain loading, a force of 0.01 N is applied on both sides of the PDMS to cause a tensile strain. The simulation result of the surface strains of the PDMS about strain is presented in Figure 2(aii). It can be predicted from the FEA results: the average particle spacing of the NP arrays on the PET will increase when the sensor is subject to pressure, while the PDMS maintain original status. The current of PET SE decreases, and the current of PDMS SE remains unchanged (Figure 2(bi)). In contrast, when the main substrate of the sensor is stressed, the current of the NP arrays on the main substrate decreases, while the electrical signal of the NP arrays on PET does not change (Figure 2(bii)).

### 2.3. Operation

The IDEs of all SEs were connected to the data acquisition (DAQ, a Keithley 2602B controlled by a computer through Labview) to measure the real-time conductance of NP arrays. We can obtain the current–time evolution curves by applying the corresponding experimental conditions. The pressure response of the NP arrays was characterized by homemade pressure response test device. This test system includes a stepper motor and an analytical balance. The NP array-based flexible multifunctional sensor was placed on the balance pan, and the stepper motor is controlled to apply pressure to the sensor. The change in the balance reading is recorded and the corresponding pressure magnitude is obtained after a simple calculation (*P* = *m* × *g*/*S*, *S* = 7.85 ×10^−5^ m^2^ is the landing area, *m* is the balance reading, *g* = 9.8 N/kg). For strain response testing, we employed an automatic strain loading apparatus managed by a stepper motor. Stepper motors are controlled by the corresponding Labview program, which can control the two sample clamps close to or far from one another to alter the strain applied on the PDMS SE. This concise operation system could help us to characterize the sensitivities, resolutions, cycle stabilities, and hysteresis of the sensor to strain and pressure, respectively.

## 3. Results and Discussion

### 3.1. Microscopic Morphological Characterization of NP Arrays

To confirm the micromorphological characteristics of deposited Pd NP arrays, scanning electron microscopy (SEM) and transmission electron microscopy (TEM) were conducted. All the micromorphological characteristic results are presented in Figure 3. First, we captured a photograph of the PET SE with the camera, and the beam spot of NPs can cover the IDEs completely (Figure 3a). Subsequently, we carried out SEM analysis of the SE at magnifications of 30 (Figure 3b), 5000 (Figure 3c), and 60,000 (Figure 3d), respectively, to evaluate the micromorphological properties of NPs more in situ. As shown in Figure 3b, the electrode structure obtained through shadow-mask evaporation remained intact, exhibiting no noticeable fractures, defects, or adhesions. This observation provides a solid foundation for the stable operation of multifunctional sensors. Additionally, in the vicinity of the silver electrodes, diffraction zones composed of nanoscale discontinuous silver films, silver islands, or silver particles were present. These zones can be regarded as part of the conductive NP arrays and do not inherently impact the performance of the SE. From Figure 3d, the NP arrays are disordered but uniformly distributed between the gap of IDEs. Clear separations exist between the NPs, and there is no agglomeration and fusion growth observed among them. Figure 3e depicts the TEM image of the Pd NP arrays. The obvious particle boundaries between the Pd NP arrays can still be seen. The NP coverage in Figure 3e is approximately 42.8%. This demonstrates that most NP arrays are separated from each other and maintain nanoscale insulating gaps, which is very advantageous for controlling electron transport between NP arrays in quantum transport manner such as tunneling or hopping [20]. The particle size statistics of the NPs in Figure 3e is shown in Figure 3f. It can be seen that the NP size follows a log-normal distribution with an average size of 12.8 nm and a full width at half maximum (FWHM) of approximately 3.2 nm. This indicates that the NP size of our sensing material is homogeneous, and the stable NP state is beneficial for the multifunctional sensor performance to remain stable.

### 3.2. Electron Transport Properties of NP Arrays

To analyze the electron transport properties, the current–voltage (*I*–*V*) curves of NP arrays on a PET substrate were measured at different temperatures. Figure 4 displays a collection of *I*–*V* curves obtained at different temperature settings. An increase in temperature enhances the steepness of the *I*–*V* curve, indicating a greater conductivity of the NP arrays. The conductance of NP arrays exhibits a positive temperature dependence, which is completely distinct from the conductance-temperature dependence in typical continuous metal films. The zero-bias conductance (${G=\frac{dI}{dV}|}_{V=0}$) of NP arrays would be chosen as the intrinsic conductance at the temperature regime from 20 K to 300 K and presented in a logarithmic form the inset of Figure 4. Notably, two distinct slopes can be observed, corresponding to two different electronic transport models: the variable range hopping (VRH) at a lower temperature regime [35,36], and the thermal activation near room temperature [37,38]. No matter what transport model corresponds to, the electronic transport in NP arrays is quantized, ensuring the high performance of sensors based on NP arrays.

### 3.3. Response Performance of Pressure-Strain Multifunctional Sensors

As the kernel of our SEs, the intrinsic response capabilities of NP arrays to different stimuli are the prerequisites of high performance in the 3D composited sensing structure. To validate these response capabilities, we initially investigated the strain response characteristics of NP arrays mounted on PDMS. Details of the specific experimental procedure is shown in the “2.3 Operation” section and Figure 5a displays a schematic of the strain response test. Figure 5b illustrates the corresponding response curves of NP arrays within a strain range of 0 to 2.7 × 10^−3^. Similar to other strain sensors [39,40], we utilized the resistance of the NP arrays, denoted as *R* = 1/*G*, to represent the output signal exclusively during strain detection. The relative resistance change (Δ*R*/*R*_0_ = (*R* − *R*_0_)/*R*_0_, *R*_0_ is the initial resistance and *R* is the resistance going a tensile strain) of the NP arrays could be characterized by the strain *ε*. As the PDMS substrate is stretched, the average distance between adjacent Pd NPs increases, leading to an overall increase in the resistance of the NP arrays (Figure 5). We denoted the strain sensitivity as gauge factor *GF* ($GF=\frac{d(\Delta R/R_{0})}{d\varepsilon }$), which is the result of the first-order derivative of the relative resistance change in the NP arrays with the corresponding strain variables. We generated plots illustrating the relative resistance change and *GF* change curves of the NP arrays under different strain conditions. With the increase in strain, the *GF* of the sensitive element exhibits a gradual rise, following a generally monotonic increasing trend except for a few isolated points. The maximum *GF* value observed within the test range for the SE is 387.63. Thanks to its remarkably high *GF*, the NP array-based multifunctional sensors can detect subtle strain as tiny as 1.34 × 10^−4^ (Figure 5c).

In fact, the excellent response of the NP arrays to strain provides itself with the potential possibility as a pressure sensor with ultrahigh sensing performance. To validate it, a NP array deposited on a PET cantilever is used to clarify the intrinsic response to pressure. Vertical pressures are applied to the overhanging side of the PET substrate. Details of the specific experimental procedure is shown in the “2.3 Operation” section and Figure 6a displays a schematic of the pressure response test. Figure 6b presents the response curve for a pressure range from 0 to 220 Pa. With the pressure loading, the root of the cantilever undergoes tensile strain (Figure 2a), resulting in a reduction in the conductivity of the NP array. The relative conductance change Δ*G/G*_0_ is applied to characterize the response of SE to pressure, where Δ*G* = *G − G*_0_, *G*_0_ and *G* denote the conductance without and with pressure loading, respectively. The sensitivity to the pressure is introduced to describe the response capability of the NP arrays as: $S_{P}=\frac{d(\Delta G/G_{0})}{dP}$. It can be observed that the relation between Δ*G*/*G*_0_ and *P* is quasi-linear, and the sensitivity can be regarded as the slope of the response curve. The sensitivity to the pressure of NP arrays is −0.223 kPa^−1^ (Figure 6b). This high sensitivity empowers our multifunctional sensors based on NP arrays to distinguish even minute pressure variations as tiny as 4.24 Pa (Figure 6c).

Figure 7 depicts the periodic response curves of the sensor for repetitive pulses of pressure and strain, respectively. In the pressure repeatability testing, a constant pressure of 130 Pa is applied to the multifunctional sensor and maintained for 15 s during each cycle, then released and kept at the baseline for another 15 s. As depicted in the figure, the NP arrays exhibit astonishingly repeatable responses to pressure throughout the test, with minimal or no response variations or baseline drift observed after multiple cycles (Figure 7a). Analogously, strain response repeatability tests were performed by loading a constant strain of 15% onto the NP arrays for 5 s in each cycle and then holding it for another 5 s after release. The NP arrays showed minimal response or baseline drift to the strain, indicating that it has strong performance repeatability (Figure 7b).

### 3.4. Multifunctional Response of Composite Sensing Materials

In spite of the excellent multifunctional precepting performance of the NP arrays, the NP array-based sensors still face the challenge of dealing with the interference between different mechanical stimuli. To investigate the reversibility of the sensing materials, the sensor was loaded and released separately with pressure and strain, and the responses of the SEs were collected simultaneously. Figure 8 provides an overview of the response behavior of the two SEs within the sensor. It has been discovered that the SE on the cantilever is not responsive to the strain, and the SE on the PDMS substrate does also insensitive to external pressure. This demonstrates the successful separation of strain and external pressure in the 3D sensing structure by confining the strain and pressure loading to distinct substrate, effectively eliminating interference between them. A response hysteresis in the sensing material is visible when the strain response return line of PDMS is observed [41]. This response hysteresis generated from the plastic deformation induced by the slip of polymer chains inside PDMS. The hysteresis for pressure response on the cantilever is significantly lower than the one for strain response on the main substrate. This is because the cantilever, with only one end fixed, can limit the generation of plastic deformation in the polymer film. It is important to note that the dispersed morphology of the NP arrays ensures minimal mechanical interaction between particles. Therefore, different from the sensing materials such as microcracking films or carbon nanotube networks [9,42], the mechanical deformation will hardly induce irreversible damage in NP arrays, thus avoiding the extra hysteresis or response drift from sensing materials. Hence, the drift (Figure 7) and hysteresis (Figure 8) in sensors come from the damage of the flexible substrates mostly. Choosing a flexible substrate with better elastic properties is expected to optimize the mechanical characteristics of sensors based on NP arrays.

Some nanomaterial-based flexible sensors, as well as this sensor, are listed in Table 1 for a comparison of their strain- and pressure-sensitive performance. Benefitting from quantum transport in nanoparticle arrays and the special mechanical structure, our sensor demonstrates superior sensing capabilities compared to other devices. Particularly, when subjected to pressure loading, the cantilever efficiently transfers the applied pressure to surface deformation, resulting in a higher sensitivity compared to other pressure-sensitive designs. Notably, the 3D composited sensing design, as demonstrated in Figure 8, offers interference-free properties, providing valuable inspiration for the development of other flexible sensing devices which are susceptible to mechanical cross-interference easily.

## 4. Conclusions

In this study, we achieved to develop a multifunctional flexible sensor which can discriminate correlative pressure and strain simultaneously. The sensor employs percolative NP arrays as the essential SEs and arranges them on the cantilever and main substrate of the 3D flexible substrate, respectively. Leveraging the high sensitivity response of the NP arrays to stimuli, our sensor exhibited a pressure sensitivity of −0.223 kPa^−1^ on the cantilever SE, with a pressure resolution of 4.24 Pa. When subjected to applied strain, within the test range the maximum *GF* value for the SE is 387.63, so that it can determine the strain as tiny as 1.34 × 10^−4^. Moreover, we also found that in NP arrays, there is almost no mechanical interaction among dispersive NPs, thus the SEs have little contribution or affection on the mechanical properties of sensor and even could decrease the hysteresis as much as possible during operation. The remarkable capabilities of our sensor can contribute to the development of intelligent and flexible devices, as well as advanced interaction methods.

## Figures and Tables

**Figure 1 sensors-23-05956-f001:**
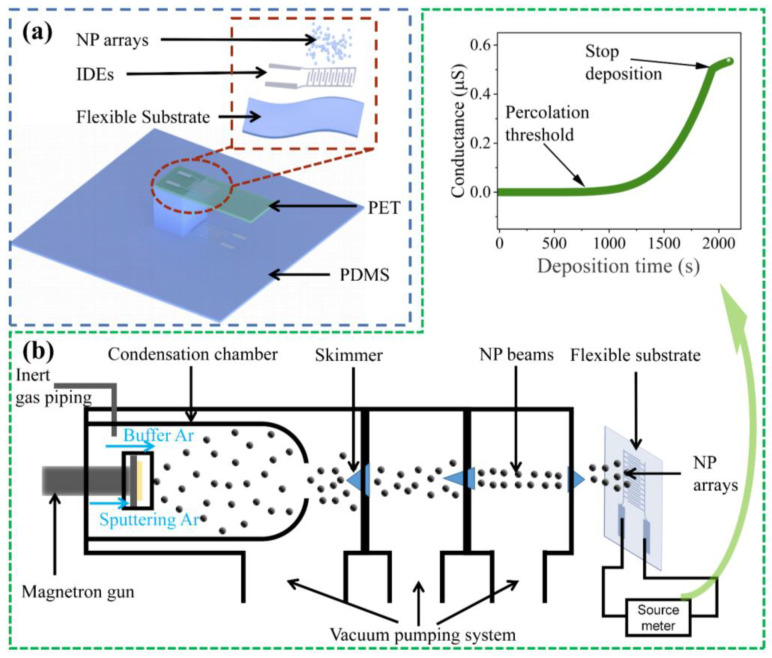
(**a**) Schematic of a NP array-based flexible multifunctional sensor; (**b**) schematic diagram of the NP deposition system and the real-time conductance evolution of NP array in deposition.

**Figure 2 sensors-23-05956-f002:**
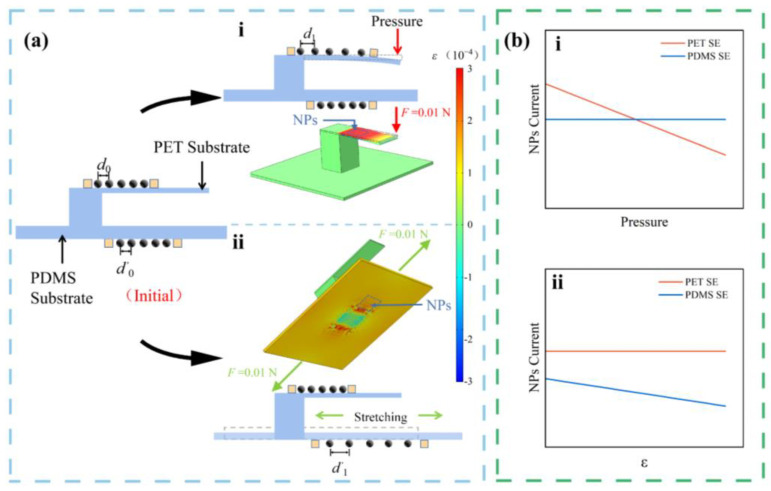
(**a**) The FEA results of (i) pressure and (ii) strain stimuli; (**b**) current evolution trends of the NP arrays responding to (i) a pressure loading and (ii) a strain on PET and PDMS.

**Figure 3 sensors-23-05956-f003:**
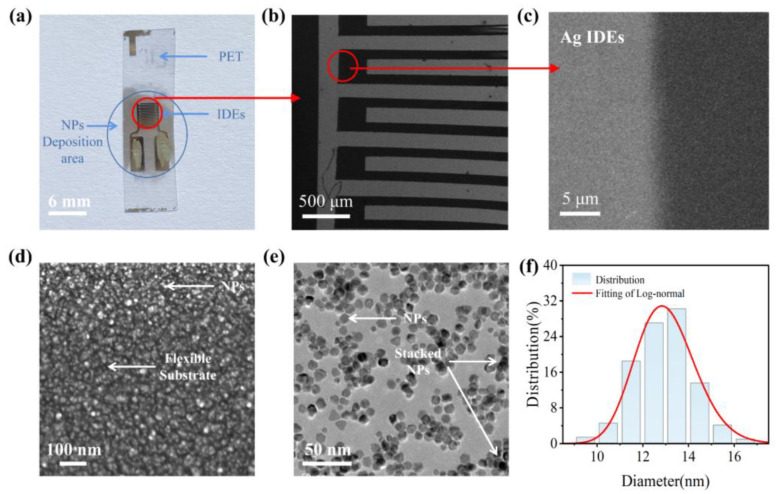
(**a**) Photograph of PET SE; (**b**) SEM image of IDEs from (**a**); (**c**) SEM image at electrode boundary; (**d**) SEM image of NPs; (**e**) TEM image of NPs; (**f**) particle size statistics of NPs.

**Figure 4 sensors-23-05956-f004:**
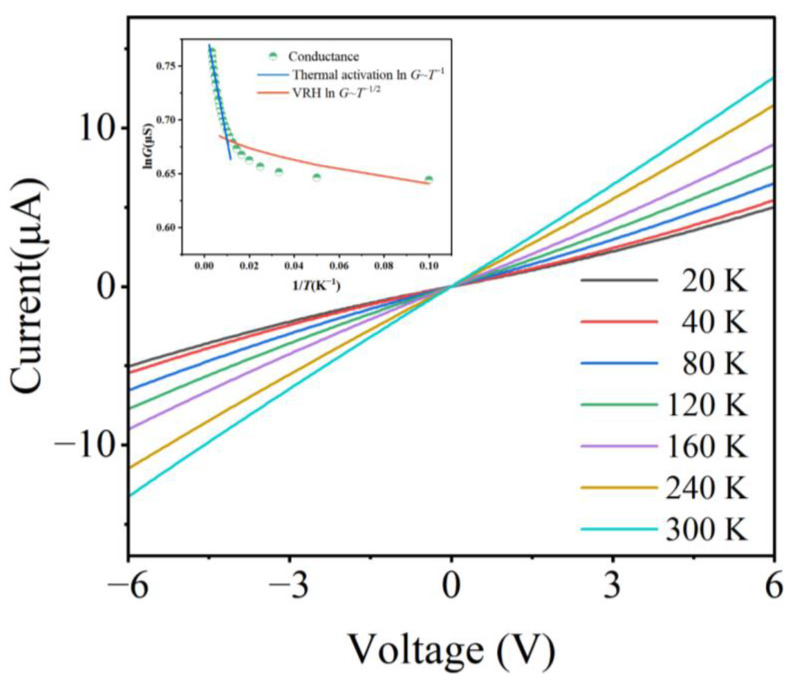
*I*–*V* curves of NP arrays at a temperature range from 20 K to 300 K (illustration represents temperature dependence of the zero-bias conductance of NP arrays).

**Figure 5 sensors-23-05956-f005:**
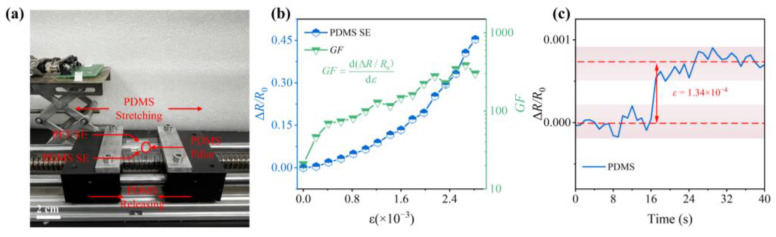
(**a**) Schematic diagram of strain response test; (**b**) resistance and *GF* change in NP arrays on PDMS under strain; (**c**) real-time response of NP arrays to the smallest detectable stain variation of 1.34 × 10^−4^.

**Figure 6 sensors-23-05956-f006:**
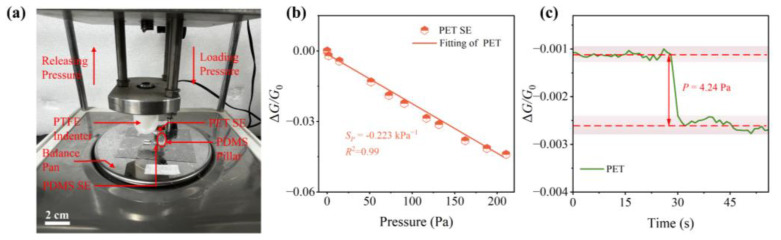
(**a**) Schematic diagram of strain response test; (**b**) conductance change in NP arrays on PET under pressure; (**c**) real-time response of NP arrays to the smallest detectable pressure variation of 4.24 Pa.

**Figure 7 sensors-23-05956-f007:**
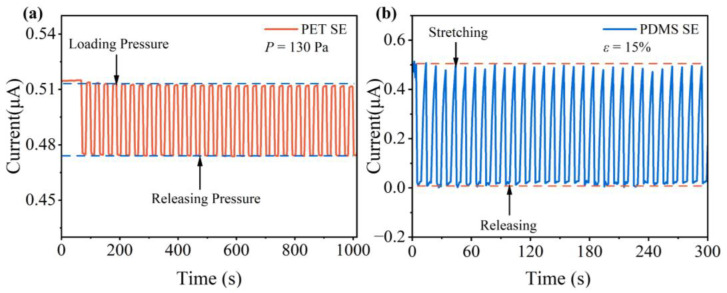
(**a**) Pressure repeatability response curve of NP arrays on PET substrate at *P* = 130 Pa; (**b**) strain repeatability response curve of NP arrays on PDMS substrate at *ε* =15%.

**Figure 8 sensors-23-05956-f008:**
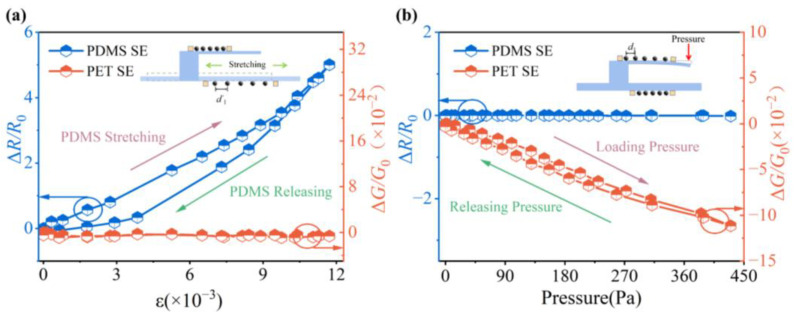
(**a**) Strain-response curves of NP arrays on PET and PDMS (includes a cycle response of loading and releasing); (**b**) pressure-response curves of NP arrays on PET and PDMS (includes a cycle response of loading and releasing).

**Table 1 sensors-23-05956-t001:** Comparison between our work and other sensors.

Sensing Materials	Sensing Structure	Strain Sensitivity (*GF*)	Strain Resolution	Pressure Sensitivity (Absolute Value)	Pressure Resolution	Existing Mechanical Interference
NPs (this work)	3D Structure	242.62	1.34 × 10^−4^	0.233 kPa^−1^	4.24 Pa	No cross-interference
NPs [28]	Membrane	35.42	—	0.13 kPa^−1^	0.5 Pa	Only applied in strain-free scenes
Carbon NPs [43]	Membrane	—	—	0.24 kPa^−1^	14 Pa	Vulnerable to strain interference
MWCNT/PDMS [44]	Membrane	0.022 kPa^−1^	—	0.026 kPa^−1^	—	Vulnerable to cross-interference
AuNW [45]	Membrane	—	—	0.08 kPa^−1^	25 Pa	Vulnerable to strain interference
CNTs [46]	Membrane	103.2	—	—	—	Vulnerable to pressure interference
AgNWs/PDMS Aerogel [47]	3D Structure	32	—	—	—	Vulnerable to pressure interference
SAA/Al^3+^/MXene Hydrogels [48]	3D Structure	4.3	—	0.075 kPa^−1^	—	Vulnerable to cross-interference
PVA/H_2_SO_4_@PU-based Sponge [49]	3D Structure	2.26	—	0.080 kPa^−1^	—	Vulnerable to cross-interference
Graphene/PDMS Sponge [50]	3D Structure	—	—	0.12 kPa^−1^	5 Pa	Vulnerable to strain interference
Hydrogel@PU Sponge [51]	3D Structure	1.33	—	0.083 kPa^−1^	—	Vulnerable to cross-interference

## Data Availability

The data that support the findings of this study are available from the corresponding.

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
