# Peer review of "A 3D Composited Flexible Sensor Based on Percolative Nanoparticle Arrays to Discriminate Coupled Pressure and Strain"

_sensors, 2023, doi:10.3390/s23135956_

Round 1

Reviewer 1 Report

Authors did a great job preparing this paper. The topic and the approach to the problem is interesting.

Please consider below comments:

·         Line 66: Add a reference for statement” because of the mechanical sensing cantilever with only one end fixed”.

·          Add more quantitative information about the sensitivity of other structures in your Introduction.

·         Line 102: Explain briefly what techniques has been used and how the purity has been measured.

·         Add how many datapoints you have in Figure 2. Do you have sufficient measurements?

·         Add an SEM image of your fabricated device.

·         Add an image of the measurement/experimental setup

English language can be improved.

Reviewer 2 Report

The authors have developed flexible sensors based on nanoparticle networks that can discriminate between the type of mechanical stimuli. It is an original work and the sensors, according to the tests, show very good sensitivities. The article is well written with enough details related to the experiments. Maybe it would be good to give some details related to the reproducibility of the method and how reliable it is. I believe that it would arouse the interest of readers in the field, perhaps due to its ability to discriminate the type of mechanical stimuli. The literature in the field abounds, but its efficiency, sensitivity and not very difficult way of preparation can bring many readers.

Reviewer 3 Report

Overall, this is a good paper. The following revisions are requested.

-Abstract does not talk about the nature of the specific research in detail.

-Former research studies accomplished in line with the performed research are not presented in detail. 

-Figure 2: Enlarge the circled areas.

-Figure 3: Label the images.

-Figure 5: Why do you use two linear curve fitting? Unjustified.

-Table 1: All compared publications are with MEMBRANE. Nothing is with 3D STRUCTURE. Why?

Round 2

Reviewer 1 Report

Thank you for revising the paper. The paper can be accepted in the current form.